# Platelet distribution width as a cost-effective marker for sepsis-associated acute kidney injury: A retrospective cross-section study

**Xuelian Yin, Jiebin Li, Enfeng Ren, Yi Zheng, Hong Zhao, Jing Zhang**\*◑**, Qingxia Du**◉\*◑

Department of Emergency, The Affiliated Tongren Hospital of Capital Medical University, Beijing, China

◑ These authors contributed equally to this work.
\* zhangjing68519@sohu.com (JZ); duqingxia2012@163.com (QD)

## Abstract

### Background

Sepsis-associated acute kidney injury (S-AKI) is a critical complication with high morbidity and mortality. The potential predictive role of platelet distribution width (PDW) in S-AKI remains to be elucidated, and its clinical implications in S-AKI are still not well understood.

### Objective

This study aims to determine whether platelet distribution width within 24 hours of admission could serve as a predictor of S-AKI in septic patients.

### Method

A retrospective analysis of platelet indices in patients with sepsis at the Affiliated Tongren Hospital of Capital Medical University, a tertiary medical center, was conducted from 2015 to 2022. Patients with sepsis were divided into two groups: an S-AKI group and a non-AKI group based on the presence of S-AKI during hospital. Clinical characteristics and laboratory parameters at admission were compared between two groups. A Multivariate logistic regression analysis was conducted to identify risk factors for S-AKI in septic patients. Additionally, receiver operating characteristics (ROC) curve was employed to evaluate the predictive value of these indices for S-AKI in septic patients.

### Result

A total of 410 patients with sepsis were included in the study, including 57 in S-AKI group and 353 in non-AKI group. The levels of PDW and average platelet volume were significantly higher in the S-AKI group compared to those in the non-AKI group. Furthermore, PDW exhibited a positive correlation with SOFA score, APACHE II

**Data availability statement:** Data cannot be shared publicly because that data contain potentially identifying or sensitive patient information. However, data are available from the Beijing Tongren Hospital Ethics Committee (contact the committee by phone: 0086-010-58268486 or E-mail: bjtrkyec@126.com./ via E-mail: zhangjing68519@sohu.com and duqingxia2012@163.com to corresponding author) for researchers who meet the criteria for accessing to potentially confidential data.

**Funding:** The author(s) received no specific funding for this work.

**Competing interests:** The authors have declared that no competing interests exist.

score, and LDH levels (r = 0.273, r = 0.153, r = 0.233), yielding *P*-values <0.001, 0.008, and < 0.001 respectively. Multivariate logistic regression analysis identified PDW (OR = 1.324, 95% CI: 1.124–1.559, *P* = 0.001), SOFA scores (OR = 1.264, 95% CI: 1.011–1.579, *P* = .040) and LDH (OR = 1.005, 95% CI: 1.002–1.008, *P* = .002) as independent risk factors for S-AKI in sepsis patients. The area under curve (AUC) values for predicting S -AKI using PDW, SOFA, LDH, and combined SOFA-PDW metrics were found to be approximately equal to 0.696 (95% CI: 0.621–0.771, *P* = .000), 0.771 (95% CI: 0.706–0.837, *P* = .000) and 0.695 (95% CI: 0.611–0.780, *P* = .000), 0.799 (95% CI: 0.739–0.858) respectively.

## Conclusion

PDW values on admission may serve as a useful potential indicator of disease severity and a potential parameter for predicting S-AKI.

## Introduction

Sepsis is a significant cause of morbidity and mortality worldwide, affecting millions of people every year and posing a serious threat to human health [1]. Platelet dysfunction plays a crucial role in multiple organ dysfunctions in sepsis. During the inflammatory response in sepsis, platelet activation and aggregation lead to the formation of microthrombi in microvessels. Widespread platelet activation and endothelial damage resulting from sepsis can exacerbate and lead to disseminated intravascular coagulation and multiple organ dysfunction [2]. Sepsis-associated acute kidney injury (S-AKI) is a frequent complication of septic patients and is an independent risk factor for mortality [3,4].S-AKI is often irreversible, making early identification of patients at risk and timely initiation of effective interventions critical in the clinical management of sepsis. Consequently, extensive studies have been conducted to identify a viable predictor for risk prediction and early diagnosis of S-AKI in the context of sepsis.

Platelet indices are components of routine blood tests that are inexpensive, quickly and easily accessible in all medical settings. These indices include platelet count, which measures the number of platelets in blood; platelet distribution width (PDW), which is elevated in the various inflammatory diseases [5]; mean platelet volume (MPV), indicative of platelet average size; and plateletcrit, which measures of volume occupied by platelets. Platelet distribution width (PDW), a measure of platelet size heterogeneity, has emerged as a promising marker of platelet activation and systemic inflammation. In septic patients, excessive accelerated platelet turnover leads to an increase in immature platelets released into peripheral blood [6]. Changes in PDW can indicate the inflammatory response status or organ dysfunction and act as a predictor of sepsis mortality [7]. However, the role of PDW in predicting S-AKI has not been elucidated, and its clinical implications in S-AKI remain unclear. The pathophysiology of S-AKI is complex, involving intricate interactions between inflammation, microvascular dysfunction, and coagulation abnormalities [8].

Given the role of platelets in these processes, we hypothesized that PDW might serve as a valuable predictor for S-AKI development in septic patients. The primary objective of this study was to investigate the relationship between PDW and S-AKI in a cohort of septic patients. Additionally, we aimed to assess the potential of PDW, alone and in combination with established clinical parameters, as a potential parameter for predicting S-AKI.

## Materials and methods

### General information

We recruited patients who were consecutively admitted for sepsis between January 2015 and January 2022 from the electronic medical record system of Beijing Tongren Hospital affiliated with Capital Medical University. The study protocol was approved by the Ethics Committee of Tongren Hospital (TREC2022-KY132). We accessed the electronic medical record system on February 1, 2023 to acquire the data required for the research purpose. Owing to our research is a retrospective observational study, participant consent was waived. During or after the data collection process, the authors anonymized personally identifiable information from participants' personal data. Adult patients (≥18 years old) diagnosed with sepsis based on the Third International Consensus Definitions for Sepsis and Septic Shock (Sepsis-3) [8] were included. Exclusion criteria (excluding those who meet any of the following criteria): ① Patients with incomplete medical history, hospitalization time ≤72 hours. ② Patients under 18 years old; ③ Patients with malignant tumor; ④ Patients with immune system diseases and hyperthyroidism; ⑤ Patients currently undergoing hemodialysis or renal transplantation, or with a history of chronic kidney disease (stage 4–5); ⑥ Individuals with acute kidney injury due to causes other than sepsis; ⑦ Patients with hematogical disorders; ⑧ Individuals using antiplatelet or anticoagulant drugs that may affect platelet and coagulation function; ⑨ Individuals with known infectious diseases (such as HIV infection, tuberculosis, etc.).

### Clinical data

The following information was recorded or calculated for all included patients: ① Age, gender. ② Combined chronic underlying diseases, including cardiovascular disease (e.g., hypertension, heart disease, arrhythmia), cerebrovascular disease (e.g., previous cerebral infarction, cerebral hemorrhage), chronic lung disease (e.g., chronic bronchitis, chronic obstructive pulmonary disease, chronic interstitial lung disease), chronic kidney disease (e.g., nephrotic syndrome, chronic nephritis); ③ Infection sites, including lung infection, abdominal infection (cholecystitis, intra-abdominal abscess, etc.), urinary tract infection, soft tissue infection, multiple site infection (defined as infections in two or more sites); ④ Acute Physiology and Chronic Health Evaluation II (APACHE II) and Sequential Organ Failure Assessment (SOFA) scores, recorded within 24 hours of admission.

### Laboratory measurements

Whole blood count data and biochemical test data of included patients within 24 hours of admission, including platelet count (PLT), plateletcrit, platelet distribution width (PDW), and mean platelet volume (MPV), procalcitonin (PCT), lactic dehydrogenase (LDH) and C-reaction protein (CRP), were extracted from the Electronic Medical Record System. Blood sample measurements were performed using certified standardized in the laboratory of the Beijing Tongren Hospital by laboratory technicians. All laboratory measurements were performed by fully automated tests and subjected to regular internal and external quality assurances procedures.

### Clinical prognostic indicators

The diagnosis of sepsis-associated acute kidney injury (S-AKI) was based on the criteria of the Kidney Disease Improving Global Outcomes (KDIGO)classification [9]: ① An absolute increase in serum creatinine of more than or equal to 0.3 mg/dl (26.4 µmol/l) within 48 hours; or ② Presenting an increase of serum creatinine to 1.5 times or more from the

baseline within 7 days or by ≥ 0.3 mg/dl within 2 days; or ③ Cumulative urine output of less than 0.5 ml/ (kg· h) for more than 6 hours. Patients were categorized into S-AKI group or the non-AKI group based on the presence of S-AKI during hospitalization.

## Statistical analysis

Statistical analysis and figure production was performed using SPSS 26.0 software. Categorical variables were presented as frequencies and proportions (%) and compared using the chi-square test. The normality of quantitative data was assessed using the Kolmogorov-Smirnov (K-S) test. Normally distributed continuous data were presented as mean ± standard deviation (SD), while non-normally distributed continuous data were represented as median with interquartile range (P25, P75). Variable selection was carried out using t-tests or non-parametric tests, as appropriate. Spearman correlation analysis was conducted using R programming language version 4.4.2. PDW was set as the dependent variable to evaluate the relationship between PDW levels and disease severity scores as well as inflammatory markers. Multivariate logistic regression analysis was performed to assess risk factors for S-AKI. Receiver Operating Characteristic (ROC) curve analysis was conducted to assess the predictive value of each candidate platelet indices for S-AKI in sepsis patients. All $P$-values were two-tailed, and the statistical significance level was set at $P < .05$.

## Results

### Comparison of clinical data between S-AKI group and non-AKI group

A total of 410 septic patients were included in the study, with a mean age of 78.8 years, and 235 (57.3%) were male. The baseline characteristics of patients stratified by the presence of S-AKI are summarized in Table 1, encompassing demographic characteristics, comorbidities and laboratory examinations. The incidence of S-AKI in sepsis was 13.9%. Patients in the S-AKI group had significantly higher SOFA score (3 [2–5] vs. 3 [2–4], $P < .001$), APACHE II score (19.70 ± 4.08 vs. 20.03 ± 3.44, $P < .001$), and Charlson index (9 [7–11] vs. 8 [7–9.75], $P < .001$) compared to those in the non-AKI group (all $P < .05$); Chronic kidney disease was more prevalent in the S-AKI group (61.4% vs. 42.5%, $P = .010$). Although the distribution of infection sites did not differ significantly ($P = .062$), respiratory infections were the most common. Laboratory measurements revealed higher levels of LDH (268 [196.3–361.5] vs. 195 [160.5–251.5], $P < .001$) and procalcitonin (0.63 [0.12–3.30] vs. 0.20 [0.06–1.09], $P = .007$) in the S-AKI group, while CRP levels showed no significant difference ($P = .065$). Levels of PDW (11.80 [10.40–14.10] vs. 11.30 [10.20–13.10], $P < .001$) and MPV (10.89 ± 6.38 vs. 10.39 ± 1.10, $P = .002$) were significantly higher in the S-AKI group ($P < .05$), while platelet count (202.21 ± 89.78 vs. 211.27 ± 89.78, $P = .001$) and plateletcrit (0.19 ± 0.09 vs. 0.21 ± 0.09, $P = .005$) were slightly lower. No significant differences were observed between the groups regarding age, gender, lactate levels, and hemoglobin.

### Multivariate logistic regression analysis of risk factors for S-AKI in septic patients

To further identify prognostic factors for S-AKI, we evaluated SOFA score, APACHE II score, and Charlson index, PCT and platelet indices using the multivariate logistic regression analysis. The results confirmed that PDW, SOFA score and LDH were confirmed to be the independent prognostic factors. Specifically, sepsis patients with higher PDW (OR = 1.324, 95% CI: 1.124–1.559, $P = .001$), SOFA scores (OR = 1. 264, 95% CI: 1.011–1.579, $P = .040$) and higher LDH (OR = 1.005, 95% CI: 1.002–1.008, $P = .002$) were more likely to develop S-AKI, as shown in Table 2.

### Correlation analysis between PDW and severity in patients with sepsis

The correlation analysis between PDW and various clinical parameters, including SOFA score, APACHEII score, Charlson's index, procalcitonin (PCT), C-reactive protein and LDH was conducted in 410 patients with sepsis, as shown in Fig 1. Spearman correlation analysis revealed a positive correlation between PDW and SOFA score (r = 0.273, $P < .001$),

**Table 1. Distribution of clinical and biological characteristics in the S-AKI and non-S-AKI groups.**

| Variable | Total (N = 410) | S-AKI (N = 57) | non-AKI (N = 353) | P value |
|---|---|---|---|---|
| Age, years | 78.8 (13.6) | 78.7 (14.3) | 78.8 (13.5) | .973 |
| Gender, n (%) | | .388 | | |
| male | 235 (57.3) | 36 (55.28) | 199 (61.33) | 0.084 |
| female | 175(42.7) | 21 (44. 72) | 154(38.67) | |
| SOFA score | 3.00 (2.00,5.00) | 3.00 (2.00,5.00) | 3.00 (2.00,4.00) | .000 |
| APACHE II score | 19.0 (17,22) | 19.70±4. 08 | 20.03±3.44 | .000 |
| Charlson score | 8.00 (7.00,10.00) | 9.00 (7.00,11.00) | 8.00 (7.00,9.75) | .000 |
| Medical history, n(%) | | | | |
| Diabetes,% | 236 (57.6) | 32 (56.1) | 204 (57.8) | .885 |
| Chronic lung disease,% | 155 (37.8) | 19 (33.3) | 136 (38.5) | .556 |
| Cardiovascular disease,% | 338 (84.3) | 48 (84.2) | 290 (82.2) | .852 |
| Cerebral disease,% | 217 (52.9) | 26 (45.6) | 191 (54.1) | .254 |
| Chronic kidney disease,% | 185 (46.1) | 35 (61.4) | 150 (42.5) | .010 |
| Infection site, n (%) | | | | .062 |
| Respiratory,% | 314 (76.6) | 40 (70.2) | 274 (77.6) | |
| Abdominal,% | 21 (5.1) | 6 (10.5) | 15 (4.2) | |
| Urinary,% | 18 (4.4) | 1 (1.8) | 17 (4.8) | |
| soft tissue,% | 9 (2.2) | 0 (0) | 9 (2.5) | |
| Mutiple,% | 46 (11.2) | 10 (17.5) | 36 (10.2) | |
| LDH,u/l | 200.0(164.0,262.5) | 268.0 (196.3,361.5) | 195.0 (160.5,251.5) | .000 |
| procalcitonin,ng/ml | 0.27(0.07,1.67) | 0.63(0.12-3.30) | 0.20(0.06-1.09) | .007 |
| CRP, mg/L | 62.50(21.05,125.45) | 91.70(35.80,148.20) | 64.30(21.29,125.15) | .065 |
| Hemoglobin,g/L | 99.57±19.89 | 99.97±20.63 | 99.55±19.77 | .104 |
| platelet count,$10^9$/L | 203.8±98.5 | 202.21±89.78 | 211.27±89.78n | .001 |
| PDW,fl | 11.6(10.3,13.4) | 11.80(10.40,14.10) | 11.30(10.20,13.10) | .000 |
| plateletcrit | 0.21±0.09 | 0.19±0.09 | 0.21±0.09 | .005 |
| MPV, fl | 10.3 (9.7,11.2) | 10.89±6.38 | 10.39±1.10 | .002 |

**Table 2. Multivariate logistic regression analysis of risk factors for S-AKI in septic patients.**

| Variable | β | Wald value | P value | OR | 95%CI Upper | Lower |
|---|---|---|---|---|---|---|
| SOFA | 0.234 | 4.232 | .040 | 1.264 | 1.011 | 1.579 |
| APACHE | 0.005 | 0.011 | .918 | 1.005 | 0.909 | 1.112 |
| Charlson's index | 0.011 | 0.021 | .885 | 1.011 | 0.870 | 1.175 |
| Chronic kidney disease | 0.303 | 0.588 | .443 | 1.354 | 0.624 | 2.936 |
| PCT | -0.005 | 0.206 | .650 | 0.995 | 0.973 | 1.017 |
| LDH | 0.005 | 9.307 | .002 | 1.005 | 1.002 | 1.008 |
| PLT | 0.010 | 1.245 | .265 | 1.010 | 0.993 | 1.028 |
| MPV | 0.037 | 0.184 | .668 | 1.038 | 0.875 | 1.232 |
| Plateletcrit | -8.582 | 0.834 | .361 | 0.000 | 0.000 | 18.946 |
| PDW | 0.280 | 11.323 | .001 | 1.324 | 1.124 | 1.559 |
| constant | -8.357 | 21.414 | .000 | 0.000 | | |

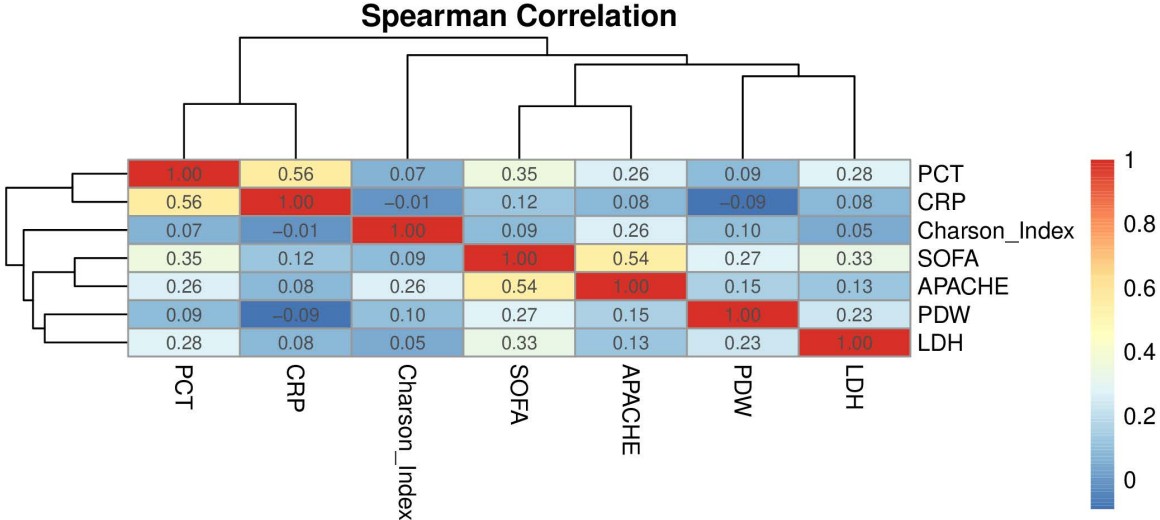

**Fig 1. Spearman's correlation matrix of PDW with disease severity scores and inflammation indicators.**

as well as APACHE II score (r = 0.153, *P* = .008). A significant correlation also existed between PDW and LDH (r = 0.233, *P* < .001). However, correlations between PDW and other markers such as Charlson's index (r = 0.084, *P* = .092), procalcitonin (PCT) (r = 0.092, *P* = .114), and C-reactive protein (CRP) (r = -0.09, *P* = .113) were not statistically significant (S1 Table).

### Efficacy of relevant indicators to predict S-AKI in septic patients

Receiver operating characteristic (ROC) curves were plotted with S-AKI group as the dependent variable and the non-AKI group as the reference group. The area-under-the curve (AUC) values for PDW, SOFA, LDH and the combined SOFA-PDW model in predicting S-AKI in septic patients were 0.696 (95% CI: 0.621–0.771, *P* = .000), 0.771 (95% CI: 0.706–0.837, *P* = .000) and 0.695 (95% CI: 0.611–0.780, *P* = .000), 0.799 (95% CI: 0.739–0.858, *P* < .001) respectively. (Fig 2) The combination of PDW and SOFA scores demonstrated superior predictive performance, with a sensitivity of 88.7% and specificity of 58.8%, and an AUC of 0.799 for S-AKI compared to SOFA scores alone. The optimal cutoff values were determined using the Youden index (sensitivity + specificity - 1). Detailed values are presented in S2 Table.

### Discussion

This study investigated the relationship between platelet distribution width (PDW) and S-AKI of 410 septic patients. Our findings suggest that PDW, along with other clinical parameters, may serve as a valuable predictor for S-AKI development in septic patients.

In our study, 13.9% of 410 septic patients developed S-AKI. This incidence is consistent with findings from a multi-center, observational study which reported that S-AKI occurs in one in six ICU patients. In the present study, patients in the S-AKI group had significantly higher SOFA score, APACHE II score, and Charlson index compared to those in the non S-AKI group (all *P* < .05). This indicates that patients with S-AKI were in a more critical condition and had more comorbidities, including a higher likelihood of chronic kidney disease. The higher prevalence of chronic kidney disease in the S-AKI group (61.4% vs. 42.5%, *P* = .010) also underscores the importance of pre-existing renal dysfunction as a risk factor for AKI in sepsis. Interestingly, our study found that PDW levels were significantly higher in the S-AKI group compared to the non-AKI group (11.80 [10.40–14.10] vs. 11.30 [10.20–13.10], *P* < .001). This finding adds to the growing body of evidence

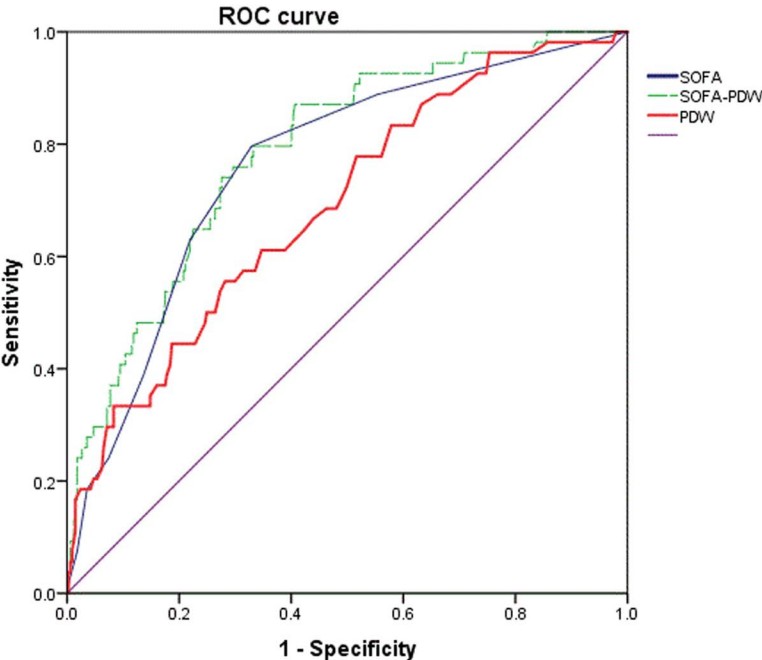

**Fig 2. ROC curves S-AKI prediction by PDW, SOFA, and SOFA-PDW.**

suggesting that alterations in platelet indices may be associated with sepsis severity and organ dysfunction [12]. The multivariate logistic regression analysis further confirmed PDW as an independent prognostic factor for S-AKI (OR = 1.324, 95% CI: 1.124–1.559, $P$ = .001), along with SOFA score and LDH levels. The observed positive correlations between PDW and various clinical severity scores—namely SOFA and APACHE II—suggest that these scoring systems reflect systemic organ dysfunction in patients with sepsis; thus, elevated PDW levels may be associated with organ dysfunction including renal impairment. PDW alongside LDH plays a role in the pathogenesis of cellular injury and acute inflammation; however, further investigation is warranted to elucidate their interplay. In summary, this study provides additional evidence supporting the potential utility of PDW as an indicator of disease severity.

Platelets play a crucial role in the inflammatory response and multiple organ failure associated with sepsis [10]. One of the primary pathological mechanisms underlying S-AKI involves platelet activation and epithelial cell damage, which lead to disrupted coagulation pathways and formation of microthrombi [11]. The formation of numerous microthrombi in patients with severe sepsis results in excessive activation and consumption of peripheral platelets, leading to thrombocytopenia. This condition triggers compensatory production of platelets in the bone marrow, resulting in an increase in immature platelets released into peripheral blood. Thrombocytopenia is also associated with organ dysfunction and increased mortality during sepsis; moreover, microthrombi may occlude renal microcirculation, causing ischemic injury that contributes to S-AKI development. In our study, we observed that platelet counts in the S-AKI group were lower than those in the non-AKI group—further confirming that more severe thrombocytopenia correlates with poorer prognosis [11].

PDW serves as a measure of platelet size heterogeneity. An elevated PDW indicates greater variation in platelet size and is often associated with increased platelet activation and turnover rates. PDW is a readily available biomarker that measures volume heterogeneity within platelet sizes while providing insights into both platelet function and morphology [12]. In the context of sepsis, platelets undergo morphological and functional changes; thus, detecting PDW can effectively reflect the status of inflammatory responses, assist in evaluating the severity of sepsis in

clinical practice, and serve as a predictive indicator for sepsis prognosis. PDW is an objective measure that reflects variability in platelet volume; its elevation indicates significant differences in platelet size due to swelling, destruction, and immaturity [6]. PDW is closely associated with platelet activation and the state of the body's inflammatory response [13].

In the context of S-AKI, elevated PDW may indicate a higher degree of platelet activation specifically related to AKI. Previous research has demonstrated that platelet indices and their kinetics are predictive markers for both sepsis [14] and mortality among patients with sepsis [15]. A study conducted by Emara M et al [16] demonstrated that patients suffering from both sepsis and AKI exhibited significantly higher levels of MPV and PDW compared to patients without AKI. Additionally, admission PDW values were associated with a more severe clinical profile along with an increased risk of 90-day mortality [17]. In our study, PDW is significantly associated with the severity of sepsis, as well as with LDH levels in septic patients, this makes PDW a valuable biomarker for predicting the intensity of inflammatory processes in sepsis. Elevated PDW values correlate with increased Sequential Organ Failure Assessment (SOFA) scores, APACHE II scores and LDH levels, all of which may contribute to clinical deterioration and mortality. Notably, PDW measured within 24 hours of admission serves as a potential parameter for predicting S-AKI.

The SOFA score is a particularly useful predictor of organ dysfunction and failure in critically ill patients [18]. LDH is routinely utilized parameter in clinical practice that is upregulated during sepsis and serve as a reliable predictor of mortality [19]. In our study, the performance of PDW in predicting S-AKI exceeded that of other more extensively studied biomarkers (e.g., PCT and LDH), and was comparable to the SOFA score. Analysis of ROC curves elucidated that PDW measured within the initial 24 hours post-admission exhibited a robust predictive capability for S-AKI, both as a standalone metric and in conjunction with SOFA scores. This implies that the PDW could facilitate rapid prognosis assessment. Clinically, it is imperative to monitor renal function meticulously in sepsis patients presenting with a PDW exceeding 12.55fl at admission, especially in those with a history of chronic kidney disease.

## Strengths and limitations

Our research presented several notable strengths. Initially, it was conducted on a medium-size study including 410 patients in Beijing, representing the general Chinese population in resource-rich cities. Subsequently, our data were meticulously extracted from our hospital's comprehensive clinical information system, thereby guaranteeing a high degree of accuracy and a minimal incidence of missing data. Moreover, the inclusion and exclusion criteria were stringent, thus eliminating individuals with conditions or under medications that might influence platelet and coagulation profiles.

Nonetheless, certain limitations are inherent to our study:(1) It is a retrospective, single center study, which may introduce selection bias; (2) The patient data span seven consecutive years, during which significant international advancements in the pathogenesis and clinical management of sepsis occurred, potentially impacting clinical outcomes; (3) The retrospective nature of AKI diagnosis introduces a degree of diagnostic bias. Consequently, the inferences drawn from this study necessitate validation through prospective investigations with substantial sample sizes.

## Conclusions

The present investigation has elucidated that PDW holds significant clinical value within the sepsis patient population, with a notable correlation being demonstrated between PDW and S-AKI. Higher PDW values were exhibited by the S-AKI cohort in contrast to the non-AKI group. Furthermore, PDW measured within the first 24 hours post-admission has been shown to be positively associated with the SOFA score, the APACHE II score and LDH levels. This finding underscores the utility of PDW as a potential biomarker for gauging the severity of inflammatory responses in sepsis. Consequently, PDW stands out as an affordable, direct and potential clinical predictive marker, which can be utilized even in settings with limited resources. Consequently, PDW emerges as a promising, straightforward, cost-effective, and efficient instrument for the assessment of S-AKI in septic patients.

## Supporting information

**S1 Fig. PDW ROC curves for predicting hospitalized mortality.**
(TIF)

**S1 Table. Spearman's correlation of PDW with disease severity scores and inflammatory markers in sepsis.**
(DOCX)

**S2 Table. PDW, SOFA, and SOFA-PDW predict ROC curves for S-AKI.**
(DOCX)

## Author contributions

**Conceptualization:** Xuelian Yin, Jing Zhang.

**Data curation:** Xuelian Yin, Jing Zhang.

**Formal analysis:** Xuelian Yin, Enfeng Ren.

**Investigation:** Qingxia Du.

**Methodology:** Qingxia Du.

**Resources:** Jiebin Li.

**Software:** Qingxia Du.

**Supervision:** Qingxia Du.

**Validation:** Jing Zhang.

**Visualization:** Yi Zheng, Hong Zhao.

**Writing – original draft:** Xuelian Yin.

**Writing – review & editing:** Xuelian Yin.

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
