## [Decision Letter · Decision Letter 0]

17 Dec 2024

PONE-D-24-44415Platelet Distribution Width as a Cost-effective Marker for Sepsis-Induced Acute Kidney Injury : a Retrospective Cohort StudyPLOS ONE

Dear Dr. Du,

Thank you for submitting your manuscript to PLOS ONE. After careful consideration, we feel that it has merit but does not fully meet PLOS ONE’s publication criteria as it currently stands. Therefore, we invite you to submit a revised version of the manuscript that addresses the points raised during the review process.

Please submit your revised manuscript by Jan 31 2025 11:59PM. If you will need more time than this to complete your revisions, please reply to this message or contact the journal office at plosone@plos.org . Please include the following items when submitting your revised manuscript:

We look forward to receiving your revised manuscript.

Kind regards,

Ennio Polilli

Academic Editor

PLOS ONE

Journal Requirements:

3. In this instance it seems there may be acceptable restrictions in place that prevent the public sharing of your minimal data. However, in line with our goal of ensuring long-term data availability to all interested researchers, PLOS’ Data Policy states that authors cannot be the sole named individuals responsible for ensuring data access (http://journals.plos.org/plosone/s/data-availability#loc-acceptable-data-sharing-methods).

Reviewers' comments:

Reviewer's Responses to Questions

**Comments to the Author**

1. Is the manuscript technically sound, and do the data support the conclusions?

Reviewer #1: No

Reviewer #2: Yes

Reviewer #3: Partly

2. Has the statistical analysis been performed appropriately and rigorously? 

Reviewer #1: No

Reviewer #2: Yes

Reviewer #3: Yes

3. Have the authors made all data underlying the findings in their manuscript fully available?

Reviewer #1: No

Reviewer #2: Yes

Reviewer #3: Yes

4. Is the manuscript presented in an intelligible fashion and written in standard English?

Reviewer #1: Yes

Reviewer #2: Yes

Reviewer #3: Yes

5. Review Comments to the Author

Reviewer #1: The authors aim to show, in a relatively large retrospective study of

410 sepsis patients, that the platelet distribution width is

predictive of sepsis-associated acute kidney injury. However, the

results do not seem to support their conclusion, as the PDW does not

seem to add a reasonably certain increase in prognostic information

compared to SOFA. Nevertheless, a more careful and detailed analysis

of this promising dataset might provide useful and new insights.

The first question that arises reading the manuscript is: why did they not

also considered the mortality of these patients, and also the timing

of clinical events, as such information, at least for in-hospital mortality,

would be immediately available and also very relevant.

Multivariate logistic analysis is mentioned in the results and the

introduction, and is the main argument of the paper, but is not

mentioned in the statistical methods.

LDH and procalcitonin and other possible indicators of S-AKI

susceptibility, especially, obviously, chronic renal disease, should

have been included in the logistic regression. While some are included

in the overall scores, elements of the scores could be more specific

for a specific prognosis, while using the scores themselves would blur

this association.

The timing between admission--when the PDW and other indices were

measured, and the AKI diagnosis should also be considered in the

analysis (possibly with a Cox model) and so should the comorbidities

that make the groups very heterogenous.

The correlation analysis should be accompanied by graphics showing the primary

data. Correlations below 0.3 are in fact very weak and also the

Pearson correlation coefficient is very sensitive to outliers,

especially if the variables do not have a Gaussian distribution.

Also, the issue of variable independence should be taken into

account. For example, the number of platelets is part of the SOFA

score, but it should be expected to be also correlated to the PDW. A

more careful examination of the individual components of the scores

and their correlation matrix could be revealing. Also, using also more

robust correlation indices, such as Spearman, might be helpful.

On page 6. the AUC for the SOFA-PDW seems to be given as 0.795 and

then, in the next paragraph, 0.799. The confidence limits also differ.

If we compare the AUC for SOFA and SOFA-PDW, that is relevant for the

supposed contribution of the PDW as a new diagnostic criterion, we see

that the difference is small (0.771 vs 0.795 or 0.799). The

confidence intervals are also mostly overlapping. The lack of an

obvious difference, that would be necessary in order to claim a

supplementary diagnostic value, is also visible in figure 1.

Thus, what the results actually seems to indicate is that the PDW does

not add significantly more prognostic information compared to SOFA,

for S-AKI, contrary to the conclusion.

At page 7, in the discussion, the authors propose a chain of

causality, sepsis -> platelet consumption -> increased PDW as an

explanation for the supposed diagnostic value of the increased

PDW for the pathologic impact of sepsis. However, a low platelet count

and increased PDW have many other causes independent of sepsis,

especially in a sample with multiple comorbidities.

Minor observations.

Page 3. `Patients with malignant tumor patients'

Page 5. `a mean age of 78.8±13.6 years' [this is the distribution not the mean]

Page 6. I think the term 'cohort' is usually employed for longer follow-up studies.

Page 8. ``ensuring data accuracy and minimal missing.''

Figure 1.

Colors in the figure are difficult to distinguish, even for a person

who does not have color visual deficiency, like myself.

Reviewer #2: Sepsis is a condition usually linked to increased risk of mortality, especially when renal failure is noticed. It is important to determine the best and adequate methods to evaluate on-time the risk of sepsis development, therefore the current work can improve the current diagnosis management and therapy response. The study was well presented, including a comprehensive description of the methodology and results. The conclusions were in accordance with the assessed data. A minor comment: the figure presenting the ROC curve should be resized (visually, it not clear enough), and considering a previous work focused on the correlation between PDW and sepsis in AKI patients which highlighted that PDW > 17.7% can associate a higher risk for AKI (including the severity of AKI), it would be nice if you could also underline (in the discussions or conclusions) the PDW value from which the risk of AKI was noticed in your patients.

Reviewer #3: I read the manuscript with great interest. This is a single-center retrospective study analyzing a potential correlation between platelet distribution width (PDW) and AKI in sepsis. The authors established well-defined study parameters, most notably exclusion criteria, clear definition of S-AKI and a specific time window of within 24 hours of admission. Statistical analysis appears sound. The correlation with several clinical severity scores was particularly interesting. There are a few opportunities for clarification/edits as outlined below.

- The overall age skewed on the high side (78+/- 13.6 years). Did the correlation between PDW and AKI differ across different age groups?

- For the exclusion criteria “individuals with acute kidney injury induced by causes other than sepsis”: how did the authors clearly and consistently delineate whether sepsis was or was not the inciting cause of AKI?

- If the inference is that increased PDW is a marker for sepsis-induced AKI, how do we know that AKI alone is not the driver for increased PDW? i.e., would we expect to see the same correlation between PDW and AKI in non-septic patients?

- Increased PDW can be caused by multiple other conditions, including respiratory and cardiovascular/cerebrovascular disorders, DM, etc. Table 1 lists comparable percentages of several such conditions for S-AKI and non-AKI groups. I would explicitly draw attention to this within the discussion section.

- Was there a sub-analysis performed to differentiate between sepsis, severe sepsis and septic shock and whether correlation between PDW and AKI was comparable or different across these categories?

- Page 8, last paragraph: “Strengths and limitations” should be a separate section.

- Page 8, last paragraph: “Large cohort “ is listed as one of the strengths, though 410 (written as 401 in that paragraph) is a medium-size study. Could the authors comment on the barriers for not including more patients?

- I agree with the authors that a seven-year study span and potential advances in sepsis diagnosis and treatment might have introduced some confounding variables. It would have been interesting to see whether PDW and AKI correlation was comparable during the first half vs second half of the study period.

- I concur that the study needs to be prospectively validated. Until then, calling PDW a marker for sepsis-induced AKI might be premature. I would reword as “potential marker” throughout the paper.

- Table 1: “Chronic kidney disease” needs a qualifier “stage 1-3”.

- There are several grammatical errors and incorrect sentence structures throughout the manuscript. E.g., Abstract, Results section, p.1: “positively correlation with the SOFA”; p.7, paragraph 2, sentence 3: “numerous microthrombi”; p.7, paragraph 2, last sentence: not grammatically correct, plus brackets with “[Error! Bookmark not defined.];” p.8, sentence 3: end of sentence missing a few words.

6. PLOS authors have the option to publish the peer review history of their article (what does this mean? ). If published, this will include your full peer review and any attached files.

**Do you want your identity to be public for this peer review?** For information about this choice, including consent withdrawal, please see our Privacy Policy .

Reviewer #1: No

Reviewer #2: No

Reviewer #3: No

---

## [Author Response · Author response to Decision Letter 1]

9 Feb 2025

Dear Reviewers,

Thank you very much for your insightful revision suggestions, which will greatly improve our research level and scientific knowledge, and enhance the scientific quality of our article. Due to the limitations of our level and the information we have received, some questions are still difficult to answer satisfactorily. If you have any questions, please contact us immediately. Thanks to the reviewer.

Kind regards,

Qingxia Du

Our response is as follows

Reviewer #1: The authors aim to show, in a relatively large retrospective study of 410 sepsis patients, that the platelet distribution width is predictive of sepsis-associated acute kidney injury. However, the results do not seem to support their conclusion, as the PDW does not seem to add a reasonably certain increase in prognostic information compared to SOFA. Nevertheless, a more careful and detailed analysis of this promising data set might provide useful and new insights.

The first question that arises reading the manuscript is: why did they not also considered the mortality of these patients, and also the timing of clinical events, as such information, at least for in-hospital mortality,would be immediately available and also very relevant.

Response:We have added information in the supplementary materials (sFig1) indicating that PDW predicts in-hospital mortality. The result was not significant, probably due to the complexity of factors influencing patient mortality, rather than being due to any single factor.

Multivariate logistic analysis is mentioned in the results and the introduction, and is the main argument of the paper, but is not mentioned in the statistical methods.

Response:We corrected and included multivariate logistic analysis within the statistical methods section.

LDH and procalcitonin and other possible indicators of S-AKI susceptibility, especially, obviously, chronic renal disease, should have been included in the logistic regression. While some are included in the overall scores, elements of the scores could be more specific for a specific prognosis, while using the scores themselves would blur this association.

Response: As shown in Table 2, logistic regression analysis indicated that neither PCT nor CKD served as risk factors for S-AKI; however, LDH demonstrated independent predictive value for S-AKI. This study suggests that molecules such as PDW and LDH may serve as potential biomarkers for predicting the occurrence of S-AKI. The factors influencing AKI are complex and multifaceted—encompassing inflammatory storms, microcirculatory dysfunctions, and cellular metabolic reprogramming. While this study highlights the predictive value of PDW concerning S-AKI, it is essential to collect more clinical case data across different disease spectra for comprehensive research purposes. The SOFA and APACHE scoring systems are widely utilized among patients with sepsis or those who are critically ill; these systems can assess multiple organ functions throughout the body comprehensively compared to traditional renal function indicators like creatinine levels and urine output.

The timing between admission--when the PDW and other indices were measured, and the AKI diagnosis should also be considered in the analysis (possibly with a Cox model) and so should the comorbidities that make the groups very heterogenous.

Response:Thank you for your insightful feedback. It would be preferable to utilize a Cox model for analysis; however, during our data retrieval process, the time variable was not considered, making it quite challenging to obtain this portion of the data again. In future studies, we will incorporate time variables in prospective designs and apply the Cox model for analysis.

The correlation analysis should be accompanied by graphics showing the primary data. Correlations below 0.3 are in fact very weak and also the Pearson correlation coefficient is very sensitive to outliers,especially if the variables do not have a Gaussian distribution.Also, the issue of variable independence should be taken into account. For example, the number of platelets is part of the SOFA score, but it should be expected to be also correlated to the PDW. A more careful examination of the individual components of the scores and their correlation matrix could be revealing. Also, using also more robust correlation indices, such as Spearman, might be helpful.

Response:We appreciate your valuable input. Based on the results of the normality tests conducted on our variables, we found that their p-values were significantly below the conventional threshold of 0.05, indicating that none of these variables adhere to a normal distribution. The specific p-values for each variable are as follows:

· PDW: 2.25E-07· Charlson Index: 3.53E-06

· SOFA: 9.03E-18

· APACHE: 4.38E-07

· PCT: 2.47E-32

· CRP: 1.20E-15

· LDH: 6.76E-32

Given that all variables failed the normality test, it is appropriate to assume they possess non-normal distributions. Consequently, I followed your recommendation and performed Spearman correlation analysis to evaluate the relationships among these variables.

The Spearman correlation matrix is presented below:

Variable PDW Charson_Index SOFA APACHE PCT CRP LDH

PDW 1 0.100 0.273 0.153 0.092 -0.092 0.233

Charlson Index 0.100 1 0.091 0.259 0.068 -0.014 0.048

SOFA 0.273 0.091 1 0.544 0.352 0.119 0.333

APACHE 0.153 0.259 0.544 1 0.259 0.077 0.126

PCT 0.092 0.068 0.352 0.259 1 0.558 0.276

CRP -0.092 -0.014 0.119 0.077 0.558 1 0.081

LDH 0.233 0.048 0.333 0.126 0.276 0.081 1

We value your suggestion and believe this approach offers a more accurate representation of the underlying structure of our data.

On page 6. the AUC for the SOFA-PDW seems to be given as 0.795 and then, in the next paragraph, 0.799. The confidence limits also differ.

Response:We checked the statistical results and made corrections.

If we compare the AUC for SOFA and SOFA-PDW, that is relevant for the supposed contribution of the PDW as a new diagnostic criterion, we see that the difference is small (0.771 vs 0.795 or 0.799). The confidence intervals are also mostly overlapping. The lack of an obvious difference, that would be necessary in order to claim a supplementary diagnostic value, is also visible in figure 1.Thus, what the results actually seems to indicate is that the PDW does not add significantly more prognostic information compared to SOFA,for S-AKI, contrary to the conclusion.

Response:Although Platelet Distribution Width (PDW) does not provide substantially more prognostic information than SOFA scores, PDW is readily obtainable and facilitates rapid prognosis assessment.

At page 7, in the discussion, the authors propose a chain of causality, sepsis -> platelet consumption -> increased PDW as an explanation for the supposed diagnostic value of the increased PDW for the pathologic impact of sepsis. However, a low platelet count and increased PDW have many other causes independent of sepsis,especially in a sample with multiple comorbidities.

Response: In sepsis, endothelial injury and tissue factor release trigger the coagulation cascade, leading to massive consumption of platelets. The platelets released by the bone marrow under stress are of varying sizes, resulting in an increase in PDW. An elevated PDW may reflect the early stage of sepsis-induced coagulopathy (SIC) or disseminated intravascular coagulation (DIC), and together with thrombocytopenia, it indicates a deterioration of the condition.

Minor observations.

Page 3. `Patients with malignant tumor patients'

Response: This has been corrected.

Page 5. `a mean age of 78.8±13.6 years' [this is the distribution not the mean]

Response: This has been corrected.

Page 6. I think the term 'cohort' is usually employed for longer follow-up studies.

Response: Yes. We have renamed our study as a retrospective cross-section study.

Page 8. ``ensuring data accuracy and minimal missing.''

Response: This has been corrected.

Figure 1.

Colors in the figure are difficult to distinguish, even for a person who does not have color visual deficiency, like myself.

Response: The ROC curve has been resized to ensure it clear enough.

Reviewer #2: Sepsis is a condition usually linked to increased risk of mortality, especially when renal failure is noticed. It is important to determine the best and adequate methods to evaluate on-time the risk of sepsis development, therefore the current work can improve the current diagnosis management and therapy response. The study was well presented, including a comprehensive description of the methodology and results. The conclusions were in accordance with the assessed data. A minor comment: the figure presenting the ROC curve should be resized (visually, it not clear enough), and considering a previous work focused on the correlation between PDW and sepsis in AKI patients which highlighted that PDW > 17.7% can associate a higher risk for AKI (including the severity of AKI), it would be nice if you could also underline (in the discussions or conclusions) the PDW value from which the risk of AKI was noticed in your patients.

Reviewer #3: I read the manuscript with great interest. This is a single-center retrospective study analyzing a potential correlation between platelet distribution width (PDW) and AKI in sepsis. The authors established well-defined study parameters, most notably exclusion criteria, clear definition of S-AKI and a specific time window of within 24 hours of admission. Statistical analysis appears sound. The correlation with several clinical severity scores was particularly interesting. There are a few opportunities for clarification/edits as outlined below.

- The overall age skewed on the high side (78+/- 13.6 years). Did the correlation between PDW and AKI differ across different age groups?

Response: This study primarily focused on cases involving elderly patients; consequently, there were relatively few instances involving middle-aged or young patients. Further research with an expanded sample size is necessary to conduct subgroup analyses based on different age demographics.

- For the exclusion criteria “individuals with acute kidney injury induced by causes other than sepsis”: how did the authors clearly and consistently delineate whether sepsis was or was not the inciting cause of AKI?

Response: Sepsis-associated acute kidney injury (S-AKI) is defined as AKI occurring within the context of sepsis.We analyzed data from an electronic medical record system where all included cases had clearly established diagnoses of sepsis and experienced AKI within seven days following their diagnosis.AKI caused by other critical illnesses such as trauma, heart failure, etc. are not included.

- If the inference is that increased PDW is a marker for sepsis-induced AKI, how do we know that AKI alone is not the driver for increased PDW? i.e., would we expect to see the same correlation between PDW and AKI in non-septic patients?

Response: I appreciate your astute observation regarding this matter; it may indeed be more accurate to characterize our inference by stating that increased PDW acts as a potential predictor for S-AKI rather than a marker.

- Increased PDW can be caused by multiple other conditions, including respiratory and cardiovascular/cerebrovascular disorders, DM, etc. Table 1 lists comparable percentages of several such conditions for S-AKI and non-AKI groups. I would explicitly draw attention to this within the discussion section.

Response: As illustrated in Table 1, when comparing individuals without AKI against those diagnosed with S-AKI, there exists a higher prevalence rate of chronic kidney disease in S-AKI was higher (61.4% vs 42.5%). However, no significant differences were noted concerning other underlying conditions such as cardiovascular or cerebrovascular diseases or diabetes mellitus across both cohorts.PDW serves as an indicator reflecting platelet activity while being intricately linked to systemic inflammation alongside coagulation status; furthermore, it has been identified as an independent risk factor associated with new cardiovascular events among Chronic Kidney Disease (CKD) sufferers.Our findings indicate that CKD does not independently predict S-AKI within septic patient populations; thus leading us to hypothesize that variations observed in levels of PDW are primarily attributable to acute inflammatory responses coupled with aberrant coagulation functions present during episodes involving septic conditions.Consequently, we posit that the level of PDW is related to acute inflammation and abnormal coagulation function in sepsis patients. PDW is an independent predictor of S-AKI and is not related to CKD.

- Was there a sub-analysis performed to differentiate between sepsis, severe sepsis and septic shock and whether correlation between PDW and AKI was comparable or different across these categories?

Response: That represents an excellent proposition. Unfortunately due constraints imposed by limited sample sizes available at hand prevented us from conducting thorough subgroup analyses focused upon varying severities inherent within selected cases.we will consider this this subject matter in future studies.

- Page 8, last paragraph: “Strengths and limitations” should be a separate section.

Response: This has been corrected.

- Page 8, last paragraph: “Large cohort “ is listed as one of the strengths, though 410 (written as 401 in that paragraph) is a medium-size study. Could the authors comment on the barriers for not including more patients?

Response: This has been corrected. 410 is the number of patients whose actual admission at our hospital meets the inclusion criteria.

- I agree with the authors that a seven-year study span and potential advances in sepsis diagnosis and treatment might have introduced some confounding variables. It would have been interesting to see whether PDW and AKI correlation was comparable during the first half vs second half of the study period.

Response: This is indeed a promising idea. We will delve into this topic in a subsequent article.

- I concur that the study needs to be prospectively validated. Until then, calling PDW a marker for sepsis-induced AKI might be premature. I would reword as “potential marker” throughout the paper.

Response:Yes, we have reworded as “potential marker” throughout the paper.

- Table 1: “Chronic kidney disease” needs a qualifier “stage 1-3”.

Response: This is a good proposition, but limited to the difficulty of reextracting the relevant data. This part of the change is more difficult, and we will consider this factor in future studies.

- There are several grammatical errors and incorrect sentence structures throughout the manuscript. E.g., Abstract, Results section, p.1: “positively correlation with the SOFA”; p.7, paragraph 2, sentence 3: “numerous microthrombi”; p.7, paragraph 2, last sentence: not grammatically correct, plus brackets with “[Error! Bookmark not defined.];” p.8, sentence 3: end of sentence missing a few words.

Response: These have been corrected.

---

## [Decision Letter · Decision Letter 1]

11 Mar 2025

Platelet  Distribution Width as a Cost-effective Marker for  Sepsis- associated  Ac ute Kidney Injury:  a Retrospective  Cross-section  Study

PONE-D-24-44415R1

Dear Dr. Du,

We’re pleased to inform you that your manuscript has been judged scientifically suitable for publication and will be formally accepted for publication once it meets all outstanding technical requirements.

Kind regards,

Ennio Polilli

Academic Editor

PLOS ONE

Additional Editor Comments (optional):

Reviewers' comments:

Reviewer's Responses to Questions

**Comments to the Author**

1. If the authors have adequately addressed your comments raised in a previous round of review and you feel that this manuscript is now acceptable for publication, you may indicate that here to bypass the “Comments to the Author” section, enter your conflict of interest statement in the “Confidential to Editor” section, and submit your "Accept" recommendation.

Reviewer #2: All comments have been addressed

Reviewer #3: All comments have been addressed

2. Is the manuscript technically sound, and do the data support the conclusions?

Reviewer #2: Yes

Reviewer #3: Yes

3. Has the statistical analysis been performed appropriately and rigorously? 

Reviewer #2: Yes

Reviewer #3: Yes

4. Have the authors made all data underlying the findings in their manuscript fully available?

Reviewer #2: Yes

Reviewer #3: Yes

5. Is the manuscript presented in an intelligible fashion and written in standard English?

Reviewer #2: Yes

Reviewer #3: Yes

6. Review Comments to the Author

Reviewer #2: You have responded to all the comments and the manuscript has been improved - it is more clearly presented the potential role of PDW in assessing S-AKI patients, and, consequently, the benefits of performing this test in clinical practice.

Reviewer #3: Thank you for your response to the reviewers' comments. The answers are satisfactory and all feedback has been addressed.

7. PLOS authors have the option to publish the peer review history of their article (what does this mean? ). If published, this will include your full peer review and any attached files.

**Do you want your identity to be public for this peer review?** For information about this choice, including consent withdrawal, please see our Privacy Policy .

Reviewer #2: No

Reviewer #3: No

---

## [Editor Report · Acceptance letter]

PONE-D-24-44415R1

PLOS ONE

Dear Dr. Du,

I'm pleased to inform you that your manuscript has been deemed suitable for publication in PLOS ONE. Congratulations! Your manuscript is now being handed over to our production team.

Kind regards,

on behalf of

Dr. Ennio Polilli

Academic Editor

PLOS ONE